# Bile Acid Malabsorption as a Consequence of Cancer Treatment: Prevalence and Management in the National Leading Centre

**DOI:** 10.3390/cancers13246213

**Published:** 2021-12-10

**Authors:** Caroline Gee, Catherine Fleuret, Ana Wilson, Daniel Levine, Ramy Elhusseiny, Ann Muls, David Cunningham, Darina Kohoutova

**Affiliations:** 1The Royal Marsden Hospital NHS Foundation Trust, Fulham Road, Chelsea, London SW3 6JJ, UK; Caroline.Gee@rmh.nhs.uk (C.G.); Catherine.Fleuret@rmh.nhs.uk (C.F.); Ana.Wilson@rmh.nhs.uk (A.W.); Daniel.Levine@rmh.nhs.uk (D.L.); Ramy.Elhusseiny@rmh.nhs.uk (R.E.); ann.muls@rmh.nhs.uk (A.M.); david.cunningham@rmh.nhs.uk (D.C.); 2St Marks Hospital, Watford Road, Harrow, Middlesex, London HA1 3UJ, UK

**Keywords:** bile acid malabsorption, cancer, diagnosis and management

## Abstract

**Simple Summary:**

Bile acid malabsorption is a common albeit a very underdiagnosed gastroenterology condition especially in cancer cohort of patients. Our study, performed at the National leading centre for treatment of cancer, focuses on prevalence and management of bile acid malabsorption in patients reviewed in our specialized clinic. Currently, precise diagnosis and excellent treatments exist for this disease.

**Abstract:**

The aim was to establish prevalence of bile acid malabsorption (BAM) and management in patients who underwent treatment for malignancy. Retrospective evaluation of data in patients seen within six months (August 2019–January 2020) was carried out. Demographic, nuclear medicine (Selenium Homocholic Acid Taurine (SeHCAT) scan result), clinical (previous malignancy, type of intervention (medication, diet), response to intervention) and laboratory (vitamin D, vitamin B12 serum levels) data were searched. In total, 265 consecutive patients were reviewed. Out of those, 87/265 (33%) patients (57 females, 66%) were diagnosed with BAM. Mean age was 59 +/− 12 years. The largest group were females with gynaecological cancer (35), followed by haematology group (15), colorectal/anal (13), prostate (9), upper gastrointestinal cancer (6), another previous malignancy (9). Severe BAM was most common in haematology (10/15; 67%) and gynaecological group (21/35; 60%). Medication and low-fat diet were commenced in 65/87 (75%), medication in 10/87 (11%), diet in 6/87 (7%). Colesevelam was used in 71/75 (95%). Symptoms improved in 74/87 (85%) patients. Vitamin D insufficiency/deficiency was diagnosed in 62/87 (71%), vitamin B12 deficiency in 39/87 (45%). BAM is a common condition in this cohort however treatments are highly effective.

## 1. Introduction

Bile acid malabsorption (BAM) was first identified in 1967 [1]. Primary biliary acids (cholic and chenodeoxycholic acids) are synthesized in 90–95% via the classic pathway in the liver from cholesterol. The synthesis involves 17 different enzymes, the rate-limiting enzyme is 7α-hydroxylase which converts cholesterol in 7α hydroxycholesterol. CYP8B1 and CYP27A1 enzymes further modificate 7α hydroxycholesterol, which generates cholic acid. Production of chenodeoxycholic acid requires CYP27A1 enzyme. Therefore, CYP8B1 expression decides about the ratio of cholic acid to chenodeoxycholic acid and CYP7A1 enzyme determines the size of the bile acid pool. Subsequently, primary bile acids are conjugated to taurine or glycine, are secreted to the bile and stored in the gallbladder. After dietary fat intake, bile acids are released to the duodenum where they play a key role in the solubilization and absorption of the lipids and fat-soluble vitamins [2,3,4,5,6].

It needs to be emphasized, that cholesterol is an important precursor of a wide range of essential molecules (not only bile acids), including steroid hormones [7]. Bjorkham et al. focused on the defective bile acid synthesis caused by the mutation in the sterol 27-hydroxylase gene (CYP27A1). Reduced bile acid synthesis leads to a compensatory increase in the activity of the rate-limiting enzyme in bile acid synthesis, cholesterol 7alpha-hydroxylase. As a consequence, 7alpha-hydroxylated bile acid precursors (especially 7alpha-hydroxy-4-cholesten-3-one) are accumulated which can result in various neurological disorders. Interestingly, activity of cholesterol 7alpha-hydroxylase can be normalized by treatment with bile acids [8,9].

The role of bile acids goes beyond being “just” a fat emulsifier: they activate different nuclear receptors (including farnesoid X-receptor) and regulate, together with insulin, glucose and lipid metabolism in the liver [10]. Bile acids serve as signalling molecules and influence epithelial cell proliferation as well as gene expression [2].

More than 95% of bile acids undergo “enterohepatic circulation”: they are reabsorbed by well characterized apical sodium-dependent bile acid transporter in the ileum (active transport) and return to the liver via the portal vein [11,12]. Consequently, synthesis of bile acids in the liver is inhibited by negative feedback regulatory pathways [2]. 

Primary bile acids are modified by intestinal microbiota: they are deconjugated and 7-α-dehydroxylation follows. This leads to formation of secondary bile acids, deoxycholic acid and lithocholic acid [13].

If enterohepatic circulation is disrupted, bile acids exert their effect towards the colon and cause chronic watery diarrhoea (through increased secretion of electrolytes and water, increased colonic permeability, increased colonic motility and increased production of mucus), abdominal bloating, rectal urgency and faecal incontinence. Patients with severe BAM can experience steatorrhoea [3,14,15,16]. 

The Selenium-Homocholic Acid Taurine (SeHCAT) scan remains the gold standard test for diagnosis of BAM [11]. The mainstay of treatment are bile acid sequestrants (colesevelam or cholestyramine) and/or dietary interventions depending on the severity of the condition [17]. 

Bile acid malabsorption can be classified into three types [18]. Type I BAM (secondary BAM) is caused by anatomically or pathologically defined enteropathy: resections of ileum, inflammatory conditions (including Crohn’s disease), chemotherapy and radiation to the organs in the pelvis [18,19]. BAM in the setting of a morphologically normal ileum (primary BAM) is classified as type II. Once thought rare [18], 33% of patients with diarrhoea-predominant irritable bowel syndrome have been found to suffer from BAM [20]. Johnston et al. confirmed that the prevalence of BAM is 1% in the general population [3]. BAM type III occurs as a consequence of gastrointestinal disorders which have not been associated with an ileal dysfunction such as previous gastric surgery (vagotomy), cholecystectomy, coeliac disease, chronic pancreatitis (with exocrine pancreatic insufficiency) and small intestinal bacterial overgrowth syndrome [11,18,20,21]. 

A significant progress in etiology/etiopathogenesis of (especially primary) BAM has been made: fibroblast growth factor 19 (FGF-19), an endocrine substance which is being released from enterocytes (upon a high intracellular bile acid level in enterocytes) into the portal circulation, activates FGF receptor 4 which subsequently leads to a downregulation of cholesterol 7α-hydroxylase, limiting enzyme of bile acid synthesis [22]. Significantly lower serum FGF-19 levels in patients with BAM were identified and an inverse relationship with serum C4 (marker of the rate of bile acid synthesis in the liver) was confirmed [23]. Defective production of FGF-19 from the ileum could be the cause of primary BAM [3]. Apical sodium-dependent bile acid transporter gene polymorphisms exist, however functional polymorphisms were also found in subjects without bile acid malabsorption and therefore Montagnani et al. concluded that the polymorphisms do not seem to affect the function of ASBT [24]. Accelerated small bowel transit bypassing ASBT receptors has been questioned in the idiopathic and post-radiotherapy cases, however this seems to be unlikely taking into account the affinity between ASBT receptor and bile acids [22,25]. Faecal micriobiota could play a role in the etiopathogenesis of BAM as Jeffery et al. reported that faecal metabolomes were able to distinguish patients with irritable syndrome with and without BAM [26].

Bile acid malabsorption is one of the most underdiagnosed conditions in gastroenterology and exclusion of this diagnosis should belong to the basic tests in patients with chronic diarrhoea [27]. 

The aim of the study was to establish the prevalence of BAM and its management in group of patients who have undergone treatment for malignancy at the Royal Marsden Hospital. 

## 2. Methods

This was a retrospective evaluation of data in patients seen within six months period (August 2019–January 2020) in a specialized clinic in a tertiary centre. 

This is a gastroenterology clinic where patients who had undergone any anti-cancer/haematology treatment in Royal Marsden Hospital, and have gastroenterology symptoms, are referred to. The most common gastroenterology symptoms of these individuals are diarrhoea, rectal urgencies, anal incontinence, abdominal bloating, severe constipation (usually chemotherapy related), weight loss, malnutrition, early satiety and rectal bleeding. 

The electronic patient records (EPR) system was searched for demographic, nuclear medicine, clinical and laboratory data.

The nuclear medicine data provided information about the result of SeHCAT scan and severity of BAM was classified accordingly (values for borderline, mild, moderate and severe BAM are: 20–15, <15, <10 and <5% of retention at seven days). 

The following clinical data was recorded: type of malignancy for which patient received oncological treatment, status of disease (remission/ongoing treatment/metastatic disease), presence of symptoms (diarrhoea, rectal urgencies, anal incontinence, abdominal bloating, weight loss), type of treatment received (radiotherapy, chemotherapy, surgery); type of intervention if diagnosis of BAM was confirmed and response to intervention. Intervention was split into three categories: medication with bile acid sequestrants, dietary guidance provided by the specialist dietitian or both. Response to treatment was confirmed as either “yes” or “no” as documented within the patients’ clinical records. See Table 1 for details. 

Laboratory data provided information about the associated vitamin deficiencies including vitamin D and vitamin B12 serum levels. These were reported at baseline, at the time of the diagnosis of BAM, before any therapeutic intervention was carried out.

The study was approved by CCR committee (number: SE 1086).

Data was treated statistically by means of descriptive statistics. 

## 3. Results

We reviewed 265 consecutive patients within six months period. Out of those, 87/265 (33%) patients (57/87 females, 66%; 30 males) were diagnosed with BAM. The mean age was 59 +/− 12 years (min. 26, max. 86 years). The largest group of patients diagnosed with BAM were females with previous gynaecological cancer (35 in total), followed by the haematology group (15), colorectal/anal (13), prostate (9), upper gastrointestinal cancer group (6) and group with another previous malignancy (9). 

Of the gynaecological group, 31/35 (89%) received radiotherapy, 26/35 (74%) chemotherapy, 24/35 (69%) surgery. A total of 14/35 (40%) females were treated with all three modalities.

Of the haematology group, 9/15 (60%) received chemotherapy; 6/15 (40%) underwent allogenic transplant and were diagnosed with gut Graft versus Host Disease (GvHD); 3/6 with an acute form of GvHD, 3/6 with chronic form of GvHD.

Of the colorectal/anal cancer group, 5/13 (38%) had colon cancer and underwent right hemicolectomy; 5/13 (38%) had rectal cancer and four out of five had undergone radiotherapy, one patient had surgery. Three patients (3/13; 24%) had anal cancer and all received radiotherapy.

The prostate group consisted of nine patients: all received radiotherapy. One individual had undergone right hemicolectomy for previous colon cancer.

Six individuals enrolled in the upper GI group underwent the following interventions: distal gastrectomy, subtotal gastrectomy, Whipple´s procedure, partial small bowel resection, laparotomy with wedge resection of the liver, ERCP with stent + palliative chemotherapy.

The group with previous malignancy consisted of four patients with history of neuroendocrine tumour (distal ileum, terminal ileum, caecum, unknown origin), two patients with invasive ductal breast carcinoma, one patient with testicular cancer, one individual with liposarcoma of the right retroperitoneum and one patient with metastatic lung adenocarcinoma.

Classification according to severity of BAM is recorded in Figure 1. 

Severe form of BAM was observed most frequently in the haematology group (10/15; 67%) and the gynaecological group (21/35; 60%) of individuals, Table 1. 

Medication and low-fat diet were commenced in 65/87 (75%) patients, medication alone was recommended to 10/87 (11%) and diet alone to 6/87 (7%) individuals. Colesevelam was used in 71/75 (95%) and cholestyramine in 4/75 patients.

No intervention followed in 6/87 (7%) patients: two suffered from constipation, symptoms of one individual resolved on morphine treatment (as a pain relief), two patients were lost to follow-up (shortly after the diagnosis of BAM) and one individual took herbal medication.

Symptoms improved in 74/87 (85%) patients, 6 individuals were lost to follow up or have not been assessed yet, symptoms of 7/87 (8%) patients have not improved. 

Vitamin D insufficiency (50–74 nmol/L) or deficiency (<50 nmol/L) were diagnosed in 62/82 (76%) of investigated patients; vitamin B12 deficiency (<239 pg/mL) was diagnosed in 39/84 (46%) of investigated patients. 

## 4. Discussion

Group of patients who have received radiotherapy for malignancy in the pelvis and/or chemotherapy are in a high-risk developing BAM.

The SeHCAT scan, regarded as the gold standard for diagnosis of BAM, was performed in 1981 for the first time [28]. The test is simple, non-invasive, well tolerated and involves minimal radiation. Currently, SeHCAT scan is available in Canada and 12 European countries [29]. In practice, the patient receives an orally administered capsule containing homotaurocholic acid labelled with 75-selenium and retention of the isotope is measured after seven days (compared to situation after three hours after the ingestion of capsule) [11,22,30,31]. 

In countries, where SeHCAT scan is not available, empirical trial (therapeutic test) can be used as a diagnostic option. Yet, improvement of symptoms may be given by placebo effect and on contrary, false negative result can be caused by poor compliance with the treatment. Furthermore, possible interactions of cholestyramine/colesevelam with other medication and side effects of bile acid sequestrants have to be taken into account [29,32]. Yet, during the COVID pandemic, we used the therapeutic test (usually with colesevelam) in a substantial number of patients even in our centre, so that multiple visits to the hospital were avoided.

Clinically, diarrhoea (>three liquid stools per day) is the most typical symptom of BAM. It has been present in all but one patient in this cohort. Rectal urgencies and anal incontinence are typical for any patients who received radiotherapy in the pelvis and/or chemotherapy, but not for those who have undergone surgery for an upper GI malignancy. Weight loss could be associated with severe form of BAM, however results have to be interpreted with caution especially in the haematology group of patients as majority of individuals from this group were still undergoing treatment.

Literature suggests that around 50% or more of patients who have received pelvic radiotherapy will develop BAM [33,34]. Female patients with history of radiotherapy for gynaecological cancer are, together with patients with history of radiotherapy for rectal cancer, individuals belonging to the especially high-risk group of individuals who can suffer from BAM. According to our experience, men after radiotherapy for prostate carcinoma do not frequently suffer from BAM unless the field for radiotherapy is larger (e.g., involves paraaortic lymph-nodes, too). Colorectal cancer patients who have undergone right hemicolectomy have also usually terminal ileum resected and can therefore develop BAM. 

Patients undergoing chemotherapy including treatment with lenalidomide for multiple myeloma can develop bile acid malabsorption [35]. We also diagnose BAM in a number of those suffering from Graft versus Host Disease, which is in agreement with other authors. The inflammatory-cell-mediated destruction of apical sodium-dependent bile acid transporters in the ileum can lead to BAM in this setting [36,37]. We observe that BAM can have transient features (especially in acute GvHD; data not shown), but can also develop into a long-term condition. 

Our results have confirmed, that if females with previous gynaecological cancer or patients with haematological diagnosis develop BAM, a severe form of BAM is observed usually (in ˃60%). 

The mainstay of treatment for BAM are bile acid sequestrants: cholestyramine, colestipol and colesevelam [17,38]. They bind the bile acids and prevent them from exerting their effect in the colon. In our practice, we use cholestyramine, which is licenced in the treatment of BAM, and colesevelam, which is being prescribed off licence at present [39]. Colesevelam is usually commenced by a gastroenterologist or specialist team, once established, general practitioners are able to take on shared care and continue prescribing locally for patients (as agreed with local clinical commissioning groups). Both, cholestyramine and colesevelam can cause side effects including constipation, headaches, abdominal pain, bloating, nausea and vomiting. From our practice, we can confirm, that colesevelam is usually better tolerated then cholestyramine, which is in agreement with literature [17,40], and therefore we tend to prefer colesevelam as the first line treatment of BAM at present. The dose should be increased gradually to avoid adverse side effects, which would limit patients’ tolerability. The required dose does not depend on severity of BAM only but can vary between individuals depending on tolerability and symptoms.

Within our clinic, patients with a diagnosis of BAM are routinely referred for a dietary assessment. They are asked to complete a 7-day dietary diary in advance of their appointment with a registered dietitian, so they receive individualised dietary advice at their first consultation. Patients are reviewed 6–8 weeks later and their diet is tailored further according to their symptoms. Low-fat diet is indicated as the first line treatment for borderline or mild BAM. Symptoms of patients with moderate or severe BAM are stabilised with colesevelam (cholestyramine) first and then the patients are reviewed by the dietitian [41]. Prospective evaluations of low-fat dietary interventions in managing BAM have demonstrated significant improvements in symptoms, where total fat has been reduced to 20% of daily energy intake [42,43]. Caution is used with patients who have a low body weight or poor oral intake and they are generally commenced on colesevelam in the first instance to optimise their symptoms. They will also be counselled on food fortification techniques and may be prescribed appropriate nutritional supplements, whilst being reviewed regularly.

If the patient suffers from remaining symptoms typical for BAM, additional supportive therapies including loperamide and/or a stool bulking agent such as sterculia can be useful.

Bile acid sequestrants have potential to bind other drugs, therefore it is recommended to take another medication either one hour before or four hours after the bile acid sequestrant [44]. 

Deficiencies of fat-soluble vitamins (A, D, E, K) can be induced by the treatment of bile acid sequestrants and serum vitamin A, D, E levels and prothrombin time are recommended to be checked periodically [44,45]. Interest in vitamin D has increased since presence of vitamin D receptors has been confirmed in many different cells. Relationship of vitamin D to immune system, bone health as well as to different conditions including cancer, cardiovascular disease, renal and liver failure, autoimmune diseases and inflammatory diseases have been investigated [46].

In our cohort we observed insufficiency/deficiency of vitamin D in 76% of investigated patients and this could even worsen after commencing treatment for BAM. We recommend the following supplementation: 1000–2000 IU vitamin D orally daily; if the deficiency is severe, we start with vitamin D 20,000 IU provided on a weekly basis. In general, vitamin D insufficiency/deficiency should be thought of in the cohort of cancer patients and investigated accordingly. Vitamin K deficiency and bleeding after long-term use of cholestyramine have been reported, albeit rarely. Yet, coagulopathy was observed not only few weeks or months after the start of the therapy, but Vroonhof reported a case of haemorrhage due to cholestyramine treatment which the patient had been on for 25 years [47]. 

Vitamin B12 (cobalamin), a vital micronutrient, water soluble vitamin, is being obtained from dietary sources. Before reaching the liver or peripheral tissues, it needs three transporters: transcobalamin I (R-protein, haptocorrin; responsible for transport from the stomach to the duodenum), intrinsic factor (produced by gastric parietal cells) and transcobalamin II. The complex of vitamin B12—transcobalamin I is split in the duodenum by pancreatic enzymes and cobalamin is bound to intrinsic factor (IF). IF is crucial for the transport (duodenum—distal ileum) as well as absorption of vitamin B12 in the distal/terminal ileum. In the setting of cancer patients, previous gastrectomy, exocrine pancreatic insufficiency, previous resections of ileum or impaired ileal function by previous pelvic radiotherapy can lead to insufficient absorption of vitamin B12. In this cohort, vitamin B12 deficiency was identified in almost half of the patients with BAM. Parenteral supplementation is required in this setting [48,49]. We recommend administering 1 mg of hydroxycobalamin on alternate days for two weeks, which is followed by a three-monthly intramuscular injection of hydroxycobalamin 1 mg. 

Prevention measures to avoid radiation induced enteropathy potentially associated with BAM are of an extreme importance, taking into account that prevalence or radiation enteropathy exceeds the prevalence of inflammatory bowel disease. Measures to prevent the development of intestinal radiation toxicity have required a substantial progress in radiation treatment planning and radiation deliver techniques over time, which has been made [50]. Use of immunomodulating drugs, enterotrophic agents and compounds with ability to modulate intraluminal content have been suggested [50]. Further, early aggressive suppression of inflammation in the terminal ileum (e.g., in Crohn’s disease) as well as early aggressive immunosuppressive treatment of Graft versus Host Disease may reduce the incidence and/or severity of BAM [51]. Nutritional strategies to prevent gastrointestinal toxicity during pelvic radiotherapy have been scrutinized. Evidence from 24 randomized controlled trials was weak for elemental, low or modified fat and low-lactose interventions and modestly positive for intervention on fibre intake during the radiotherapy. Probiotics seem to be promising, potentially, however further studies are required [52]. Recently, Jeffery et al. looked at the faecal microbiome in patients with and without irritable bowel syndrome in those with and without BAM. Significant differences in faecal metabolomes were identified and the authors concluded that this new knowledge could be used in the development of microbe-based treatments [26]. We can therefore hypothesize, that this approach could be considered as a prevention measure, too. 

## 5. Conclusions

Bile acid malabsorption is a common condition observed in the cohort of cancer patients. Excellent diagnostic tool and effective treatment exist at present. After appropriate intervention, symptoms and quality of life can be improved significantly. 

## Figures and Tables

**Figure 1 cancers-13-06213-f001:**
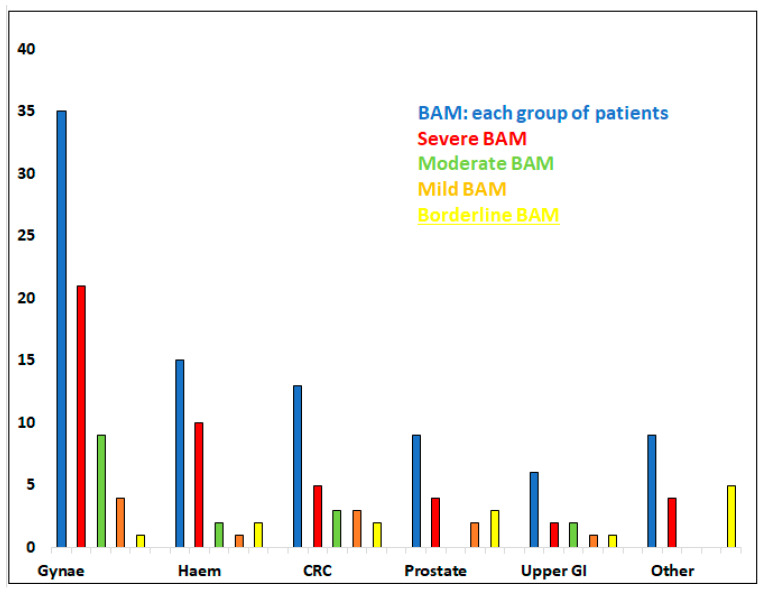
Patients with previous malignancy diagnosed with BAM (87 in total). Split into groups depending on original diagnosis: patients with gynaecological malignancy (Gynae; 35 patients); haematological diagnosis (Haem; 15); colorectal/anal cancer (CRC; 13); prostate (9); upper gastro-intestinal malignancy (upper GI; 6) and other (9). Further, divided into subgroups according to severity of BAM (severe, moderate, mild, borderline BAM).

**Table 1 cancers-13-06213-t001:** Clinical and radionuclide data.

	Status of Disease	Diarrhoea	Rectal Urgencies	Anal Incontinence	Abdominal Bloating	Weight Loss	Type of Treatment	Severity of BAM	Type of Intervention	Response to Intervention
	Remission: 1						Radiotherapy: R	Mild or borderline: 1	Diet: D	
	Ongoing treatment: 2						Chemotherapy: C	Moderate: 2	Medication: M	
	Metastatic disease: 3						Surgery: S	Severe: 3		
Gynecology group	1: 33/35 (94%)	35/35 (100%)	33/35 (94%)	31/35 (89%)	25/35 (71%)	14/35 (40%)	R +/− C +/− S: 30/35 (86%)	1: 5/35 (14%)	D-M: 29/35 (83%)	30/35 (86%)
	2: 1/35 (3%)							2: 8/35 (23%)	D: 1/35 (3%)	
	3: 1/35 (3%)							3: 22/35 (63%)	M: 2/35 (5%)	
									3/35 (9%): no intervention	
Haematology group	1: 4/15 (27%)	15/15 (100%)	11/15 (73%)	6/15 (40%)	12/15 (80%)	4/15 (27%)	C +/− R +/− transplant: 13/15 (87%)	1: 3/15 (20%)	D-M: 13/15 (87%)	15/15 (100%)
	2: 11/15 (73%)							2: 1/15 (7%)	M: 2/15 (13%)	
								3: 11/15 (73%)		
CRC/anal group	1: 11/13 (85%)	12/13 (92%)	11/13 (85%)	8/13 (62%)	7/13 (54%)	2/13 (15%)	C-S: 5/13 (38%)	1: 6/13 (46%)	D-M: 11/13 (85%)	12/13 (92%)
	2: 1/13 (8%)						R-C: 4/13 (31%)	2: 3/13 (23%)	D: 1/13 (8%)	
	3: 1/13 (8%)						R-S: 2/13 (15%)	3: 4/13 (31%)		
							R-C-S: 1/13 (8%)			
							C: 1/13 (8%)			
Prostate group	1: 7/9 (78%)	9/9 (100%)	9/9 (100%)	7/9 (78%)	8/9 (89%)	1/9 (11%)	R-S: 5/9 (56%)	1: 5/9 (56%)	D-M: 5/9 (56%)	8/9 (89%)
	3: 2/9 (22%)						R: 3/9 (33%)	3: 4/9 (44%)	M: 3/9 (33%)	
							S: 1/9 (11%)		1/9 (11%): no intervention	
Upper GI group	1: 4/6 (67%)	6/6 (100%)	1/6 (17%)	1/6 (17%)	3/6 (50%)	1/6 (17%)	S: 4/6 (67%)	1: 1/6 (17%)	D-M: 5/6 (83%)	6/6 (100%)
	2: 2/6 (33%)						C-S: 1/6 (17%)	2: 3/6 (50%)	M: 1/6 (17%)	
							C: 1/6 (17%)	3: 2/6 (33%)		

## Data Availability

Data is contained within the article.

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
