# Peer review of "Bile Acid Malabsorption as a Consequence of Cancer Treatment: Prevalence and Management in the National Leading Centre"

_cancers, 2021, doi:10.3390/cancers13246213_

Round 1

Reviewer 1 Report

I have read the response of authors to most of my comments  and I believe the manuscript has been significantly improved and now warrants publication in Cancers. 

Reviewer 2 Report

The paper Is very original and interesting. It Is amenable to read.

This manuscript is a resubmission of an earlier submission. The following is a list of the peer review reports and author responses from that submission.

Round 1

Reviewer 1 Report

This is a nicely presented descriptive study regarding the incidence of BAM in cancer patients treated in a tertiary cancer center. 

Although, data are quite well presented, the study overall is limited to the description of BAM incidence. It would be nice to present more detailed data regarding the outcome of these patients per type of cancer and per sweverity of the disease, along with analysis regarding factors that predict response to treatment. Furthermore, there is no detailed presentation of the clinicopathological characteristics of these patients as well as the status of the disease (eg. initial diagnosis, relapse, extent of metastatic disease etc) that could better describe the vulnerable patient population. In addition, treatments and outcome should be correlated by type of treatment. 

No statistics section exists in the methods. 

Description of the study center is of no value in the introduction section. In case authors consider this is of importance, it should be included in the methods. 

Author Response

To the reviewer 1

Thank you for your kind comments. Please, see our responses below.

Although, data are quite well presented, the study overall is limited to the description of BAM incidence. It would be nice to present more detailed data regarding the outcome of these patients per type of cancer and per severity of the disease, along with analysis regarding factors that predict response to treatment. Furthermore, there is no detailed presentation of the clinicopathological characteristics of these patients as well as the status of the disease (eg. initial diagnosis, relapse, extent of metastatic disease etc) that could better describe the vulnerable patient population. In addition, treatments and outcome should be correlated by type of treatment. 

Thank you for this comment. We have prepared a comprehensive table (Table 1) in the revised version. As the response to intervention was ˃86 % in any group tested, further analysis in regards of factors which predict response to treatment is not possible and outcome cannot be correlated with type of treatment used.

We have added this text into the paper:

Methods:

The following clinical data was recorded: type of malignancy for which patient received oncological treatment, status of disease (remission / ongoing treatment / metastatic disease), presence of symptoms (diarrhoea, rectal urgencies, anal incontinence, abdominal bloating, weight loss), type of treatment received (radiotherapy, chemotherapy, surgery); type of intervention if diagnosis of BAM was confirmed and response to intervention. Intervention was split into three categories: medication with bile acid sequestrants, dietary guidance provided by the specialist dietitian or both. Response to treatment was confirmed as either “yes” or “no” as documented within the patients´clinical records. See Table 1 for details.

Discussion:

Clinically, diarrhoea (˃ three liquid stools per day) is the most typical symptom of BAM. It has been present in all but one patient in this cohort. Rectal urgencies and anal incontinence are typical for any patients who received radiotherapy in the pelvis and/or chemotherapy, but not for those who have undergone surgery for an upper GI malignancy. Weight loss could be associated with severe form of BAM, however results have to be interpreted with caution especially in the haematology group of patients as majority of individuals from this group were still undergoing treatment.

No statistics section exists in the methods. 

Thank you for this comment.

We have added this paragraph into the section “Methods“:

Data was treated statistically by means of descriptive statistics.

Description of the study center is of no value in the introduction section. In case authors consider this is of importance, it should be included in the methods. 

Thank you.

We have moved the description of the study centre from the “Introduction“ to “Methods“.

Professor Darina Kohoutova

Reviewer 2 Report

The authors should provide more information about the patients: presenting complaints, bowel habits and gastrointestinal symptoms, weight loss or gain, status of neoplasm after treatment.

What kind of "specialized clinic" do you run? The only explanation given is that it looks "solely after patients who have undergone treatment for malignancy." Is it a gastroenterology clinic? It seems odd that so many of the patients would receive SeHCAT studies unless they were selected for having diarrhea or other gastrointestinal problems. The type of setting will allow readers to decide how their own patients might compare to yours.

If space is at a premium, the treatment part of the Discussion could be abbreviated. Also, Figures 2 and 3 don't add much to presenting the information in a sentence or two.

Author Response

To the reviewer 2

Thank you for your kind comments. Please, see our responses below.

The authors should provide more information about the patients: presenting complaints, bowel habits and gastrointestinal symptoms, weight loss or gain, status of neoplasm after treatment.

Thank you for this comment. We have prepared a comprehensive table (Table 1) and have added this text into the paper:

Methods:

The following clinical data was recorded: type of malignancy for which patient received oncological treatment, status of disease (remission / ongoing treatment / metastatic disease), presence of symptoms (diarrhoea, rectal urgencies, anal incontinence, abdominal bloating, weight loss), type of treatment received (radiotherapy, chemotherapy, surgery); type of intervention if diagnosis of BAM was confirmed and response to intervention. Intervention was split into three categories: medication with bile acid sequestrants, dietary guidance provided by the specialist dietitian or both. Response to treatment was confirmed as either “yes” or “no” as documented within the patients´clinical records. See Table 1 for details.

Discussion:

Clinically, diarrhoea (˃ three liquid stools per day) is the most typical symptom of BAM. It has been present in all but one patient in this cohort. Rectal urgencies and anal incontinence are typical for any patients who received radiotherapy in the pelvis and/or chemotherapy, but not for those who have undergone surgery for an upper GI malignancy. Weight loss could be associated with severe form of BAM, however results have to be interpreted with caution especially in the haematology group of patients as majority of individuals from this group were still undergoing treatment.

What kind of "specialized clinic" do you run? The only explanation given is that it looks "solely after patients who have undergone treatment for malignancy." Is it a gastroenterology clinic? It seems odd that so many of the patients would receive SeHCAT studies unless they were selected for having diarrhea or other gastrointestinal problems. The type of setting will allow readers to decide how their own patients might compare to yours.

Thank you for this question. We have added the explanation into the text:

This is a gastroenterology clinic where patients who had undergone any anti-cancer / haematology treatment in Royal Marsden Hospital are referred to. The most common gastroenterology symptoms of these individuals are: diarrhoea, rectal urgencies, abdominal incontinence, abdominal bloating, severe constipation (usually chemotherapy related), weight loss, malnutrition, early satiety and rectal bleeding,

If space is at a premium, the treatment part of the Discussion could be abbreviated. Also, Figures 2 and 3 don't add much to presenting the information in a sentence or two.

Thank you for this.

As the information from Figure 2 is included in Table 1 now, we have omitted Graph 2. We also have omitted Graph 3. 

Professor Darina Kohoutova

Reviewer 3 Report

I have now reviewed the Manuscript ID; cancers-1379589

I think the subject is very interesting and perhaps has been neglected for some time. I appreciate the efforts that were put into this research work. I have the following comments and issues to improve the manuscript.

  1. The introduction is good in highlighting the role and the importance of bile acids through out the body. It should contain a section on the interaction of bile acids synthesis and metabolism with glucocorticoids.
  2. Vit D Deficiency/Insufficiency should receive great attention and its vital role to the multiple systems of the body.
  3. Please discuss the status of Glucocorticoids (GCs) in these patients and their effects in relation to BAM. What is the role of 11B-HSD1 in relation to 7alpha-hydroxylase and bile acids metabolism?
  4. What was the prevalence of BAM in the general populations compared to cancer patients?
  5. Is there a need for figure 2 (data are mentioned in figure 1)? Instead, the reader would like a figure on the interaction between bile acids and GCs synthesis and metabolism.
  6. Why the treatments did not include Vit D supplements and in some Vit B12 intake?
  7. How long  Vitamin D insufficiency/deficiency has been in these patients without monitoring?
  8. How valid the results of Vit D and Vit B12 stated in figure 3 have been? The authors did not mention how these data were obtained and on what time scale?
  9. Conclusion should also recommend the monitoring of bile acids in these patients as well as monitoring and follow up of Vit D status in cancer patients as a whole.

Author Response

To the reviewer 3

Thank you for your kind comments. Please, see our responses below.

I think the subject is very interesting and perhaps has been neglected for some time. I appreciate the efforts that were put into this research work. I have the following comments and issues to improve the manuscript.

1. The introduction is good in highlighting the role and the importance of bile acids through out the body. It should contain a section on the interaction of bile acids synthesis and metabolism with glucocorticoids.

Thank you for your kind comment.

Still, we are not convinced that interaction of bile acid synthesis and metabolism with glucocorticoids would be of (clinical) relevance and importance in the context of this clinical study.

Further, majority of patients did not have treatment with glucorticoids at the time of the diagnosis of BAM.

2. Vit D Deficiency/Insufficiency should receive great attention and its vital role to the multiple systems of the body.

Thank you, we have discussed this more in the section “Discussion“.

3. Please discuss the status of Glucocorticoids (GCs) in these patients and their effects in relation to BAM. What is the role of 11B-HSD1 in relation to 7alpha-hydroxylase and bile acids metabolism?

Thank you for this comment, but the patients have not been on treatment with glucocorticoids unless it was part of treatment – of the ongoing chemotherapy (subgroup of haematology group of patients only). We have not investigated any serum glucocorticoids in these patients.

4. What was the prevalence of BAM in the general populations compared to cancer patients?

We have not had our control group of individuals, however Johnston et al. (ref 3) acknowledged that the prevalence of BAM is 1 % in general population

We have added this into the text.

5. Is there a need for figure 2 (data are mentioned in figure 1)? Instead, the reader would like a figure on the interaction between bile acids and GCs synthesis and metabolism.

Thank you. We have omitted Figure 2. We are not convinced, that the interaction of bile acids and the synthesis of glucocorticoids would be relevant in the context of the study.

6. Why the treatments did not include Vit D supplements and in some Vit B12 intake?

Thank you for this comment.

Patients who had their serum vitamin B12 and vitamin D levels investigated (at the time of BAM diagnosis) have not been on any supplementation of vitamin D and/or vitamin B12 so far.

The following statement is included in the “Methods“:

… “These were reported at baseline, at the time of the diagnosis of BAM, before any therapeutic intervention was carried out“….

We have added into the text the recommended supplementation:

…. We recommend the following supplementation: 1000-2000 IU vitamin D orally daily; if the deficiency is severe, we start with vitamin D 20000 IU provided on a weekly basis“.

…”We recommend to administer 1mg of hydroxycobalamin on alternate days for two weeks, which is followed by a three-monthly intramuscular injection of hydroxycobalamin 1mg”.  

7. How long Vitamin D insufficiency/deficiency has been in these patients without monitoring?

Thank you. We cannot provide an honest answer as majority of patients who had this investigated at the time of the diagnosis of BAM have never had this investigated.

8. How valid the results of Vit D and Vit B12 stated in figure 3 have been? The authors did not mention how these data were obtained and on what time scale?

Thank you.

The data were obtained at the time of the diagnosis of BAM.

We have included the following statement into the text:

“These were reported at baseline, at the time of the diagnosis of BAM, before any therapeutic intervention was carried out“

9. Conclusion should also recommend the monitoring of bile acids in these patients as well as monitoring and follow up of Vit D status in cancer patients as a whole.

Thank you, we disagree with the recommendation that monitoring of bile acids is necessary. The diagnosis of BAM has to be though of in this cohort of patients which is emphasized in the text.

We have emphasized though that the vitamin D status in cancer patients should be monitored and followed up in the cancer cohort patients in general.

Professor Darina Kohoutova

Round 2

Reviewer 2 Report

None.

Reviewer 3 Report

I have now carefully read the revised version of the above manuscript and I am happy that the authors have improved to certain extent. However, I feel that my concerns were not fully answered.  I would like some mention in the introduction of the etiology of BAM and what are the measures that have to be taken to avoid the condition in the future within the discussion. In addition, the discussion should include some scientific info to the readers on the pathway of bile acids synthesis from cholesterol in relation to other steroid hormones explaining the cause of BA deficiency in addition to malabsorption.

It is important to inject some science in to this weak paper. The 7α-hydroxylase cholesterol (CYP 7A1) is a key enzyme of this catabolic pathway that determines the size of the bile acid pool, catalyzing the hydroxylation of cholesterol to 7αhydroxycholesterol (Björkhem et al., 2010).